# Noiser: Bounded Input Perturbations for Attributing Large Language Models

**Mohammad Reza Ghasemi Madani**[1]* **Aryo Pradipta Gema**[2] **Gabriele Sarti**[3]
**Yu Zhao**[2] **Pasquale Minervini**[2,4] **Andrea Passerini**[1]
[1]University of Trento   [2]University of Edinburgh   [3]CLCG, University of Groningen
[4] Miniml.AI

## Abstract

Feature attribution (FA) methods are common post-hoc approaches that explain how Large Language Models (LLMs) make predictions. Accordingly, generating faithful attributions that reflect the actual inner behavior of the model is crucial. In this paper, we introduce NOISER, a perturbation-based FA method that imposes bounded noise on each input embedding and measures the robustness of the model against partially noised input to obtain the input attributions. Additionally, we propose an *answerability* metric that employs an instructed judge model to assess the extent to which highly scored tokens suffice to recover the predicted output. Through a comprehensive evaluation across six LLMs and three tasks, we demonstrate that Noiser consistently outperforms existing gradient-based, attention-based, and perturbation-based FA methods in terms of both faithfulness and answerability, making it a robust and effective approach for explaining language model predictions.[1]

## 1 Introduction

Transformer-based language models (Vaswani et al., 2023) are fundamental to the latest advancements in natural language processing (Team et al., 2024; Touvron et al., 2023; Bai et al., 2023; DeepSeek-AI, 2025; OpenAI et al., 2024). However, they are often perceived as opaque (Rudin, 2019; Doshi-Velez & Kim, 2017; Lipton, 2018), sparking significant interest in the development of algorithms that can automatically explain the behavior of these models (Denil et al., 2015a; Sundararajan et al., 2017a; Camburu et al., 2018; Rajani et al., 2019; Luo et al., 2022).

Feature attribution (FA) techniques are popular post-hoc methods that generate token-level importance scores to highlight the contribution of each token to a prediction (Denil et al., 2015b; Jain et al., 2020; Kersten et al., 2021). The top-$p$% important tokens are typically considered as the prediction rationale (Zaidan et al., 2007; Sundararajan et al., 2017b; DeYoung et al., 2020). The quality of a rationale is often evaluated using *faithfulness* metrics, which measure to what extent the rationales accurately reflect the downstream task on model predictions.

Perturbation-based FA methods aim to explore neural networks by modifying the input of a model and observing the changes in the output to indicate which parts of the input are particularly important for inference. These methods are widely adopted in computer vision, leveraging the continuous nature of image inputs, where localized noise or masking preserves semantic coherence and avoids distribution shifts (Ivanovs et al., 2021). In contrast, NLP models face inherent challenges due to the discrete structure of the text, where even minor perturbations—whether token substitutions or embedding modifications—can push inputs out-of-distribution (OOD), destabilizing predictions and confounding attribution analysis (Liu et al., 2019).

---

*Correspondence to: rezamadani.ai@gmail.com.
[1]Our code: https://github.com/qasemii/Noiser

This divergence underscores the need for bounded perturbations in NLP, ensuring perturbed inputs remain in-distribution. Our work bridges this gap by exploring noise thresholds that alter token embeddings while preserving the original prediction to limit perturbation-induced OOD issues. Particularly, we introduce a perturbation-based FA by exploring a model's robustness against *noisy inputs*—examples created by introducing small alterations to the input embeddings without changing the model's original prediction—enabling reliable explanations grounded in the model's trained operational domain while quantifying feature importance through robustness to controlled perturbations. Our work makes the following contributions:

- We empirically show that NOISER is consistently more faithful than nine popular FAs by conducting comprehensive experiments, covering three tasks and six LMs of varying sizes from three different model families;
- We propose a new plausibility metric, *answerability*, which measures the extent to which the top-$p\%$ attributed input tokens sufficiently support the target output. By leveraging language models, this metric assesses whether a minimal subset of input tokens is adequate for generating the expected prediction, providing a simulatable alternative to human plausibility judgments.

## 2 Background

### 2.1 Generative Language Modeling

In generative language modeling, the input consists of a sequence of tokens, denoted as $X = [x_0, \ldots, x_{T-1}]$. The objective is to develop a model, $\mathcal{F}\theta$, that estimates the probability distribution $P$ over the token sequence $X$. In this context, $\mathcal{F}\theta$ represents a specific pre-trained generative language model characterized by parameters $\theta$.

$$P(x_0, \ldots, x_{T-1}) = \mathcal{F}_\theta(x_0) \prod_{t=1}^{T-1} \mathcal{F}_\theta(x_t \mid x_0, \ldots, x_{t-1})$$

### 2.2 Input Importance for Generative LMs

Given a model $\mathcal{F}_\theta$, our objective is to determine the importance distribution of the input tokens for each predicted token $x_T$, based on the preceding sequence $X = [x_0, \ldots, x_{T-1}]$. A feature attribution method, denoted as $e_T$, applied at position $T$, yields an importance distribution $S_T = [s_0, \ldots, s_{T-1}]$ corresponding to the target token $x_T$, where a higher value of $s_i$ indicates greater importance of the input token $x_i$ in predicting $x_T$.

$$e_T(\mathcal{F}_\theta, X, x_T) \rightarrow S_T$$

### 2.3 Bounded Perturbations

Bounded perturbations refer to small, structured uncertainties in mathematical systems where a specified constraint limits the perturbation magnitude. These are critical for analyzing system robustness against disturbances while ensuring predictable behavior.

Perturbation is a minor alteration to a system, such as $\delta$ added to a nominal matrix $A$, resulting in $A + \delta$. This captures uncertainties or disturbances. If the perturbation magnitude is bounded as $\|\delta\| \leq \epsilon$, where $\epsilon > 0$, it is called a bounded perturbation. Consider the nominal system $\dot{x} = Ax$. Under a perturbation $\delta$, the system becomes:

$$\dot{x} = (A + \delta)x.$$

## 3 Our Method

Let $\mathbf{n} \in \mathbb{R}^{d_{\text{model}}}$ denote a noise vector where each component $n_i \sim \mathcal{N}(0, 1)$. We form noisy examples from original inputs by imposing small perturbations to the input embeddings,

such that the noisy input results in the model outputting an incorrect answer. For this purpose, we first pass a prompt $X = [x_0, \ldots, x_{T-1}]$ into $\mathcal{F}_\theta$ to collect the probability distribution, $P$, over the model's vocabulary with $x_T$ being the most likely output (*i.e.*, $\mathcal{F}_\theta(X) = x_T$).

In the next step, we utilize a *binary search* algorithm to find the **maximum** scaling factor $k$ such that if we perturb the embedding of a targeted token with $\mathbf{n}_{\text{scaled}} = k \cdot \mathbf{n}$ the model wouldn't change its initial prediction $x_T$. Specifically, we set $x_i := x_i + \mathbf{n}_{\text{scaled}}$ and let $\mathcal{F}_\theta$ to continue, giving us a set of corrupted probabilities $P^*_{x_i}$. Because $\mathcal{F}_\theta$ partially loses information about the corrupted token, the probability of $x_T$ from the first step would likely be lower in $P^*_{x_i}$.

We repeat the process for each token until we obtain $K = [k_0, \ldots, k_{T-1}]$ where each $k_i$ is the maximum scaling factor such that if we corrupt the embeddings of $x_i$ using $\mathbf{n}_{\text{scaled}} = k_i \cdot \mathbf{n}$, the model wouldn't change its original output. The mathematical representation of $K$ is illustrated below:

$$K = \{k_i \mid \forall k > k_i \Rightarrow \mathcal{F}_\theta(X^i_{\text{perturbed}|k}) \neq x_T, \mathcal{F}_\theta(X^i_{\text{perturbed}|k_i}) = x_T\}, \quad i \in \{0, \ldots, t-1\}$$

where $X^i_{\text{perturbed}|k} = [x_0 \ldots (x_i + \mathbf{n}_{\text{scaled}}) \ldots x_{t-1}]$ is the input sequence in which $x_i$ is altered with $\mathbf{n}_{\text{scaled}} = k \cdot \mathbf{n}_{\text{bounded}}$. The equation above indicates that each scale factor $k_i$ is such that for all values $k$ greater than $k_i$ if we perturb $x_i$ using $\mathbf{n}_{\text{sclaed}} = k \cdot \mathbf{n}_{\text{bounded}}$ to create a noisy input $X^i_{\text{perturbed}|k}$, $\mathcal{F}_\theta$ would return a different output from the original one ($x_T$).

In the final step, we find the $k_{\min} = \min(K)$ to generate the final noise samples $\mathbf{n}_{\text{scaled}} = k_{\min} \cdot \mathbf{n}$ to add to each token embedding and obtain the token scores using the following:

$$S = \{s_i \mid s_i = p(X) - p(X^i_{\text{perturbed}|k_{\min}})\}, \quad i \in \{0, \ldots, t-1\}$$

Using $k_{\min}$, we ensure to perturb the input enough to reach a flipping point in prediction to get the minimal set of features needed to achieve this outcome. The intuition is that tokens with higher importance are more sensitive to noise injection, resulting in a larger reduction in the model's output likelihood.

To show the effectiveness of selecting $k_{\min}$, we propose different boundings and measure their faithfulness. We analyse **i)** using maximum noise across tokens ($k_{\max}$); **ii)** individual token maximum noise where $k$ is different for each input token and is the maximum the model can tolerate ($k_{\max}$ per token); **iii)** norm-bounded setting where the noise vector $\mathbf{n}$ is divided by the expected value of the noise vector $L_p$ norm, $\mathbb{E}\left[\|\mathbf{n}\|_p\right]$; and **iv)** random $k$ where $k$ is randomly selected from the uniform distribution. The details of each configuration are provided in Section 5.

## 4 Experiment

### 4.1 Model & Data

In our study, we employ variants of Qwen (Bai et al., 2023), Gemma (Team et al., 2024), and Llama (Touvron et al., 2023) models. We choose our models to span from hundreds of millions to a few billion parameters as we want to explore how the model size affects the faithfulness of each FA. All models used are publicly available. [2]

We use KNOWN dataset[3] provided by Meng et al. (2023) and LONG-RANGE AGREEMENT (LONGRA; Vafa et al., 2021) to conduct our analysis. Besides, for long generation we utilize WIKIBIO (Lebret et al., 2016). The following is an instance from the KNOWN dataset.

---

[2]We use checkpoints from the Huggingface library for each model.
[3]Dataset can be found at: `https://rome.baulab.info/data/dsets/known_1000.json`

*LeBron James professionally plays the sport of* [**basketball**]

The LongRA dataset consists of word pairs that exhibit either a semantic or syntactic relationship. Additionally, Vafa et al. (2021) incorporate a distractor sentence, which provides no relevant information about the word pair, to evaluate long-range agreement. An example from the LongRA dataset is shown below, with the distractor included in parentheses.

*When my flight landed in **Japan**, I converted my currency and slowly fell asleep. (I had a terrifying dream about my grandmother, but that's a story for another time). I was staying in the **capital**,* [***Tokyo***]

WIKIBIO is a dataset consisting of Wikipedia biographies. We use the first two sentences as a prompt, similar to Manakul et al. (2023). The model is then expected to continue generating the biography. This task is inherently more open-ended compared to the previous two.

*Super Mario Land is a 1989 side-scrolling platform video game*

The computation of each task's faithfulness is provided in Section 4.5.

## 4.2 Baselines

Following previous works, we compare our rationalization method to a variety of gradient- and attention-based baselines (Vafa et al., 2021). **Input×Gradient** (Denil et al., 2015b) uses embedding gradients multiplied by the embeddings; **Integrated Gradients** (Sundararajan et al., 2017b) integrate overall gradients using a linear interpolation between a baseline input (all zero embeddings) and the original input. **Gradient SHAP** (Lundberg & Lee, 2017) compute the gradient w.r.t. randomly selected points between the inputs and a baseline distribution; **DeepLIFT** (Shrikumar et al., 2019) compares the activation of each neuron to its 'reference activation' and assigns contribution scores according to the difference. **Sequential Integrated Gradients** (Enguehard, 2023) extends Integrated Gradients by breaking down the input perturbation into sequential steps, computing gradients at each step, and aggregating them to provide more stable and interpretable attributions, **Last Attention** (Jain et al., 2020) uses the last-layer attention weights averaged across heads; **Attention Rollout** (Abnar & Zuidema, 2020) recursively computing the token attention in each layer, e.g., computing the attention from all positions in layer $l_i$ to all positions in layer $l_j$, where $j < i$; **LIME** (Ribeiro et al., 2016) trains a linear surrogate model using data points randomly sampled locally around the prediction. **Occlusion** (Zeiler & Fergus, 2014) involves systematically occluding different portions of the input and observing the impact on the output confidence.

## 4.3 Faithfulness Metrics

To assess whether a rationale extracted with a given FA is faithful, *i.e.,* actually reflects the true model reasoning (Jacovi & Goldberg, 2021), various faithfulness metrics have been proposed (Arras et al., 2017; Serrano & Smith, 2019; Jain et al., 2020; DeYoung et al., 2020). Sufficiency and comprehensiveness (DeYoung et al., 2020) are two widely used metrics that effectively capture rationale faithfulness (Chrysostomou & Aletras, 2021; Chan et al., 2022b). Both metrics use a hard erasure criterion for perturbing the input by entirely removing (*i.e.,* comprehensiveness) or retaining (*i.e.,* sufficiency) the rationale to observe changes in predictive likelihood. This hard criterion ignores the importance of each individual token, treating them all equally for computing sufficiency and comprehensiveness.

We evaluate rationales using soft sufficiency (Soft-NS) and comprehensiveness (Soft-NC) proposed by Zhao & Aletras (2023) to measure the faithfulness of the full importance distribution. Using these metrics, instead of entirely removing or retaining tokens from the input, we randomly mask parts of the token vector representations proportionately to their FA importance. The summation of Soft-NC and Soft-NS is considered as the final faithfulness score. For the detailed implementation of these metrics, please refer to Appendix A.

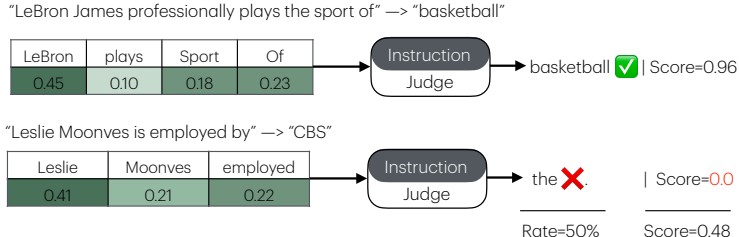

Figure 1: Answerability metrics evaluation. To get the answerability metrics, the judge model is instructed to predict the completion token given a limited set of tokens from the original prompt.

## 4.4 Answerability Metrics

The utilization of LLMs has emerged as a prominent trend across numerous research domains (Peng et al., 2023; Zhou et al., 2023a; Taori et al., 2023). With any given instructions, LLMs are expected to generate responses that align with these instructions (Chen et al., 2024; Li et al., 2024; Xu et al., 2023; Longpre et al., 2023).

This capability, known as the "instruction following" ability, serves as a key metric for assessing the effectiveness of LLMs (Chen et al., 2024; Zhao et al., 2024; Taori et al., 2023; Zheng et al., 2023). To facilitate a more thorough assessment, several benchmarks have been introduced with a focus on instruction following (Zhou et al., 2023b; Qin et al., 2024).

We exploit this progress in our answerability metric by framing attribution evaluation as an instruction-based completion task. To see whether the attributions illustrate any meaningful association with a predicted output, we extend our evaluation of FAs through prompt engineering. For this purpose, we aggregate the attribution scores of each word's sub-tokens to derive word-level scores. Then we select the top-$p$% most important words w.r.t. their scores and feed these words as input to a judge model along with a task prompt. The task prompt asks the judge model to predict the completion token using this limited set of words. Feldhus et al. (2023) offer a complementary perspective—while they leverage LLMs for generating interpretability-enhancing verbalizations, our approach instead uses an LLM to directly quantify whether the selected tokens are sufficient for the prediction task.

We evaluate attributions by computing the number of samples for which the judge model generates the correct output, which we define as the FAs *answerability rate*. Additionally, for these correctly predicted samples, we aggregate the word-level attribution scores to obtain the so-called *answerability score*. For both metrics, higher values indicate better performance. In our evaluation, a higher answerability rate means that a larger proportion of samples allow the LM judge to correctly predict the output using only the minimal set of tokens. Likewise, a higher answerability score—reflecting a greater aggregated attribution mass within that token set—indicates that the attribution method is more effective at isolating the minimal semantic requirements for prediction.

This evaluation pipeline specifically applies to datasets such as KNOWN where the gold label is a meaningful word that must be inferred from the input sequence. Figure 1 shows an answerability evaluation example. The prompt used for this evaluation is shown in Appendix E.

## 4.5 Implementation Details

None of our experiments involved training or fine-tuning any language models. All FAs are built upon Inseq library (Sarti et al., 2023; 2024) except Last Attention and Attention Rollout, which we used the codebase from Zhao & Shan (2024). For NOISER, we generate 10 different noise vectors during the corrupted run for more consistent results. Binary search is done in 10 steps, yielding the accuracy of $\approx 0.001$ for the scaling factor ($k$). For KNONW and

KNOWN

| Method | Qwen2-0.5B | Llama3.2-1B | Qwen2-1.5B | gemma-2-2B | gemma-2-9B | Llama3-8B | Average |
|---|---|---|---|---|---|---|---|
| Last Attn | -0.0857 | -0.0601 | 0.0607 | 0.1407 | -0.2788 | -0.0095 | -0.0388 |
| Rollout | -0.1161 | -0.0471 | 0.1211 | 0.4624 | -0.3607 | -0.2625 | -0.0338 |
| SHAP | 0.4946 | 0.3746 | 0.5390 | 0.3726 | 0.9203 | 0.1925 | 0.4823 |
| IxG | 0.2117 | 0.7059 | 0.4612 | 0.5233 | 1.0276 | 0.5891 | 0.5865 |
| IG | 0.2176 | 0.5428 | 0.5163 | 0.2015 | 1.0355 | 0.3759 | 0.4816 |
| DeepLIFT | 0.3030 | 0.5473 | 0.5323 | 0.3557 | 0.8638 | 0.5174 | 0.5199 |
| SIG | 0.0361 | 0.3534 | 0.3003 | -0.1879 | 0.7877 | 0.2755 | 0.2609 |
| LIME | 0.2439 | 0.5103 | 0.3826 | 0.3567 | 0.4832 | 0.6555 | 0.4392 |
| Occlusion | 0.1627 | 0.5373 | 0.2477 | 0.5341 | 0.5221 | 0.7831 | 0.4645 |
| **NOISER** | **2.1854** | **1.3989** | **1.4400** | **1.4433** | **2.1767** | **2.2175** | **1.8103** |

LONGRA

| Method | Qwen2-0.5B | Llama3.2-1B | Qwen2-1.5B | gemma-2-2B | gemma-2-9B | Llama3-8B | Average |
|---|---|---|---|---|---|---|---|
| Last Attn | 1.9148 | 0.3255 | -0.0110 | -0.2382 | -0.2382 | 1.0762 | 0.4715 |
| Rollout | 1.8517 | 0.2451 | 0.0802 | -0.2643 | -0.2643 | 1.2283 | 0.4794 |
| SHAP | 3.7970 | 1.2837 | 1.6276 | 1.9746 | 2.2769 | 0.7696 | 1.9549 |
| IxG | 3.8972 | 1.7299 | 1.5370 | 2.5803 | 2.5803 | 2.0796 | 2.4007 |
| IG | 4.3388 | 1.3066 | 1.5498 | 1.3023 | 1.3023 | 3.7190 | 2.2531 |
| DeepLIFT | 4.4991 | 1.7889 | 1.5512 | 2.7428 | 2.7428 | 2.1258 | 2.5751 |
| SIG | 3.8645 | 0.9272 | 1.1047 | 0.5412 | 0.5412 | 1.1618 | 1.3568 |
| LIME | 1.0765 | 0.2212 | -0.4147 | -0.1636 | -0.1636 | 2.2995 | 0.4759 |
| Occlusion | 3.9424 | 1.9887 | 1.0145 | 3.4418 | 3.4418 | 4.2240 | 3.0089 |
| **NOISER** | **6.8055** | **4.8072** | **3.1779** | **4.2727** | **6.1681** | **5.1627** | **5.0657** |

WIKIBIO

| Method | Qwen2-0.5B | Llama3.2-1B | Qwen2-1.5B | gemma-2-2B | gemma-2-9B | Llama3-8B | Average |
|---|---|---|---|---|---|---|---|
| Last Attn | 1.0605 | 0.6304 | -0.7054 | -0.2579 | 0.2815 | 0.5500 | 0.2598 |
| Rollout | -0.6404 | 0.5591 | -0.7066 | 0.5085 | 0.3498 | 0.8785 | 0.1582 |
| SHAP | 1.4702 | 1.1672 | 1.1213 | 0.7966 | 3.1494 | 1.4063 | 1.5185 |
| IxG | 3.4273 | 1.8365 | 1.3942 | 1.5816 | 2.6047 | 1.3747 | 2.0365 |
| IG | 2.4216 | 1.5797 | 0.6975 | 1.1909 | 4.1117 | 0.6876 | 1.7815 |
| DeepLIFT | 3.2207 | 1.6265 | 1.4590 | 1.4607 | 2.3006 | 1.2739 | 1.8903 |
| SIG | 3.7656 | 1.4300 | 2.0816 | 1.4256 | 5.2280 | 1.3620 | 2.5488 |
| LIME | 3.0009 | 0.5656 | 1.1714 | 0.7180 | 2.9527 | 0.8349 | 1.5406 |
| Occlusion | 5.1051 | 2.0019 | 3.8916 | 2.7232 | 4.9300 | 3.3885 | 3.6734 |
| **NOISER** | **8.7624** | **3.7385** | **4.9864** | **4.2527** | **7.1509** | **4.6089** | **5.5833** |

Table 1: Faithfulness scores across tasks.

LONGRA datasets, where the models must provide a single output, we filter down samples to the ones that the model can correctly generate the gold output. See Appendix C for the details. For WIKIBIO, we generate 10 tokens for input completion. To obtain the faithfulness score in this task, we compute the faithfulness of each next token w.r.t. all the previous tokens and consider the averaged score as the final faithfulness. The judge model used to get the answerability metrics is `Llama-3.3-70B-Instruct-Turbo`. We chose the top-50% of the most important words from the input prompt to get the answerability score and rate.

## 5 Results

Table 1 presents the faithfulness scores across different tasks. Following Zhao & Shan (2024), each score is computed as the logarithm of the ratio between the method's score and the random baseline. Consequently, scores below zero indicate less faithful methods than the random baseline, *i.e.,* unfaithful. As shown in Table 1, faithfulness varies across different FA methods and generative models. Notably, NOISER consistently achieves higher faithfulness scores across all tasks and models, outperforming traditional FAs. This suggests that NOISER provides more reliable attributions, reinforcing its effectiveness in evaluating model faithfulness.

To demonstrate the effectiveness of selecting $k_{\min}$, we compare its faithfulness performance against alternative bounding strategies across different models in Table 2. The results show

| Scaling Factor ($k$) | Qwen2-0.5B | Llama3.2-1B | Qwen2-1.5B | gemma-2-2B | gemma-2-9B | Llama3-8B | Average |
|---|---|---|---|---|---|---|---|
| random $k$ | 1.1519 | 1.0993 | 0.7165 | 1.2287 | 1.6007 | 1.4522 | 1.2082 |
| None ($k=1$) | 1.0922 | 1.0219 | 0.6445 | 1.1844 | 1.4726 | 1.1070 | 1.0871 |
| $\mathbb{E}\left[\|\mathbf{n}\|_2\right]^{-1}$ | 1.4849 | 1.3031 | 1.2617 | 1.7236 | 2.7905 | 1.7470 | 1.7185 |
| $\mathbb{E}\left[\|\mathbf{n}\|_\infty\right]^{-1}$ | 0.8989 | 0.9515 | 0.7164 | 1.1275 | 1.4300 | 1.4115 | 1.0893 |
| $k_{\max}$ per token | 1.2230 | 0.9938 | 0.5850 | 1.2897 | 1.8359 | 1.0984 | 1.1710 |
| $k_{\max}$ | 1.0962 | 1.0203 | 0.6515 | 1.1824 | 1.4753 | 1.1059 | 1.0886 |
| $k_{\min}$ | **2.1854** | **1.3989** | **1.4400** | 1.4433 | 2.1767 | **2.2175** | **1.8103** |

Table 2: Comparison of different boundings on the faithfulness score on KNOWN dataset.

| Method | Qwen2-0.5B | | Llama3.2-1b | | Qwen2-1.5B | | gemma-2-2b | | gemma-2-9b | | Llama3-8b | | Average | |
|---|---|---|---|---|---|---|---|---|---|---|---|---|---|---|
| | Rate | Score | Rate | Score | Rate | Score | Rate | Score | Rate | Score | Rate | Score | Rate | Score |
| Last Attn | 14% | 0.0936 | 48% | 0.2496 | 10% | 0.0670 | 39% | 0.2064 | 37% | 0.1854 | 39% | 0.2081 | 31% | 0.1684 |
| Rollout | 8% | 0.0527 | 13% | 0.0649 | 8% | 0.0557 | 9% | 0.0457 | 22% | 0.1033 | 27% | 0.1364 | 16% | 0.0812 |
| SHAP | 22% | 0.1805 | 29% | 0.1890 | 24% | 0.1862 | 17% | 0.1249 | 11% | 0.0764 | 26% | 0.1454 | 22% | 0.1504 |
| IxG | 27% | 0.2177 | 33% | 0.2412 | 26% | 0.1942 | 35% | 0.2408 | 35% | 0.2537 | 30% | 0.2079 | 31% | 0.2259 |
| IG | 20% | 0.1638 | 28% | 0.1827 | 18% | 0.1426 | 16% | 0.1197 | 12% | 0.0875 | 27% | 0.1360 | 20% | 0.1387 |
| DeepLIFT | 21% | 0.1753 | 34% | 0.2279 | 26% | 0.1991 | 32% | 0.2225 | 30% | 0.2107 | 31% | 0.2019 | 29% | 0.2062 |
| SIG | 21% | 0.1583 | 21% | 0.1271 | 20% | 0.1520 | 9% | 0.0617 | 26% | 0.1978 | 12% | 0.0627 | 18% | 0.1266 |
| LIME | 37% | 0.2986 | 25% | 0.1692 | 41% | 0.3308 | 45% | 0.3003 | 50% | 0.3291 | 36% | 0.2423 | 39% | 0.2784 |
| Occlusion | 53% | 0.3689 | **49%** | 0.3223 | **54%** | **0.4224** | **48%** | 0.3152 | **52%** | 0.3323 | **50%** | 0.3726 | **51%** | 0.3556 |
| **NOISER** | **55%** | **0.5063** | 37% | **0.3665** | 43% | 0.4099 | 43% | **0.4102** | 49% | **0.4497** | 41% | **0.4858** | 45% | **0.4381** |

Table 3: Answerability metrics on KNOWN dataset w.r.t. judge model top-1 predition.

that $k_{\min}$ consistently yields the highest faithfulness scores, confirming its superiority in preserving model behavior under noise perturbation.

Since $k_{\min}$ is model-dependent, we introduce norm-bounding as a flexible alternative, where the noise vector $\mathbf{n}$ is scaled based on the model's embedding size (see Appendix D). We further compare our approach with $k_{\max}$, which applies the maximum $k$ across all input tokens ($\max(K)$), and a variant, $k_{\max}$ per token, which applies a per-token maximum scaling factor. The latter performs slightly better, as it results in a less aggressive perturbation than the global $k_{\max}$, reducing the likelihood of extreme changes in model behavior.

Additionally, we analyze the effects of unbounded scaling ($k=1$) and random $k$, where $k$ is sampled from a uniform distribution for each input sample. The consistently lower faithfulness scores in these settings highlight the necessity of proper bounding strategies to maintain faithfulness.

Overall, $k_{\min}$ is the only configuration that guarantees the model does not change its prediction under noise, making it the most reliable choice. The detailed computation of expected norm values is provided in Appendix D.

In addition to faithfulness, we monitored the runtime efficiency of NOISER and compared it with other FA methods. As expected, attention-based methods were the fastest, followed by gradient-based methods. On the other hand, perturbation-based techniques are generally more computationally demanding. However, the runtime of NOISER can be further reduced by decreasing the number of binary search steps (e.g., from 10 to 5), which yields a noticeable speed-up with only a slight trade-off in faithfulness.

The *answerability* rate and score are reported in Table 3. While NOISER achieves the highest answerability score in most cases and on average, Occlusion attains the highest answerability rate. This indicates that when NOISER attributions are deemed answerable (rate), the importance scores assigned to the top-$p$% tokens are significantly high, which is desirable. In contrast, Occlusion produces a higher number of answerable attributions but with lower scores, implying that it does not assign as much weight to key tokens.

To provide a more flexible analysis of answerability metrics, we examine cases where the gold prediction appears within the top-5 predictions of the judge model. Under this evaluation, the gap between NOISER's answerability score and those of other baselines widens, while its answerability rate also improves and approaches that of Occlusion, which achieves the best rate.

To further contextualize the answerability metrics, we also analyzed the answerability rate when all tokens (top-100%) are provided to the judge model. This represents an upper bound on the judge model's performance, offering insight into how often the judge model itself can recover the gold prediction using the full input. In our case, using all tokens, the judge model achieves approximately 84% accuracy when considering only the top-1 prediction, and around 92% accuracy when allowing for the gold output to appear among the top-5 predictions. This highlights the intrinsic limitations of the judge model and provides a reference point when interpreting answerability rates obtained from top-$p$% subsets.

Another aspect regarding the FA methods' efficiency that we evaluate is their ability to identify the minimal set of tokens most relevant to the output. We visualize the importance scores assigned by each method to critical tokens in the LONGRA "country-capital" category. Additionally, we examine the distribution of importance scores across the distractor and main parts in Figure 2. As shown in Figure 2, NOISER assigns the highest importance to critical tokens while effectively disregarding the distractor section, demonstrating a stronger focus on the main part compared to the best-performing baselines.

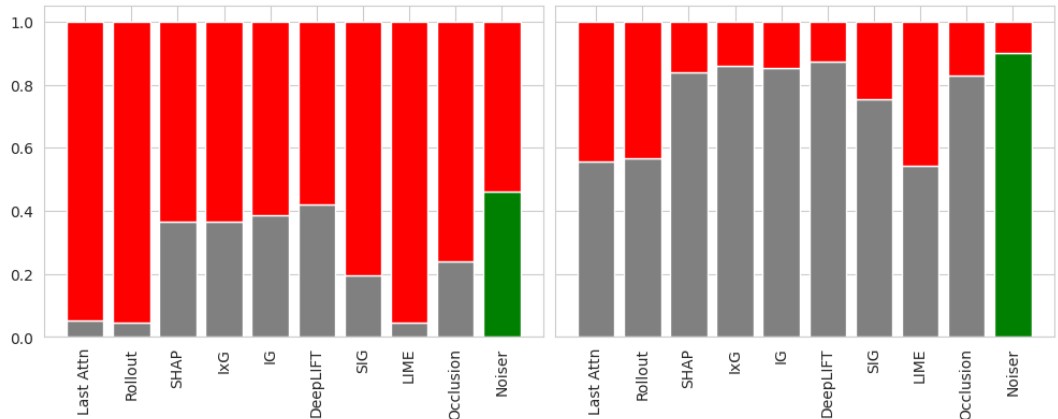

Figure 2: The aggregated score that each FAs put different parts of inputs from the "capital-world" subclass in LONGRA dataset. The red indicates the score assigned to the undesired part (e.g., distractor). The left image illustrates the aggregated score on "country"+"capital" token. The right image indicates the overall score on the main part against the distractor.

Finally, we analyzed the minimum proportion of top attributions required for each FA method to ensure that the judge model correctly predicts the original output. To determine this value, we first computed attributions for each sample using a given FA method. Then, starting with the full set of tokens, we iteratively removed the least important tokens one by one until the judge model produced an incorrect prediction. We repeated this process across all samples, averaging the proportion of retained tokens to obtain the final minimum top-$p$% required for accurate prediction. A lower value indicates a more effective FA method in identifying the most relevant attributions. In this regard, Occlusion requires the least number of tokens overall, which aligns with the results in Table 3, while Lime and NOISER take the second and third best place with minimal difference.

## 6 Related Works

Post-hoc explanation methods, such as FA techniques, are applied retrospectively by seeking to extract explanations after the model makes a prediction. Most FAs have been proposed in the context of classification tasks, where a sequence input $X = [x_0, \ldots, x_{t-1}]$ is associated with a true label $y$ and a predicted label $\hat{y}$. The underlying goal is to identify which parts of the input contribute more toward the prediction $\hat{y}$ (Atanasova et al., 2020; Wallace et al., 2020; Madsen et al., 2022; Chrysostomou & Aletras, 2022; Lei et al., 2016; Chan et al., 2022a; Ghasemi Madani & Minervini, 2023). Most FAs generally fall into gradient, attention, and perturbation-based categories.

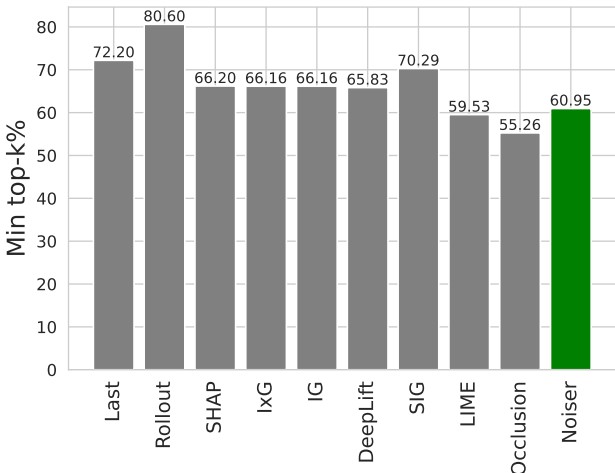

Figure 3: Minimum top-$p$% attribution required for the judge model to retain the correct prediction across different feature attribution methods. Lower values indicate higher attribution accuracy, as fewer tokens are needed to maintain the original output.

| Dataset | Input | Output |
|---|---|---|
| Known | LeBron James professionally plays the sport of | basketball |
| LongRA | When my flight landed in Japan , I converted my currency and slowly fell asleep . ( I had a terrifying dream about my grandmother , but that ' s a story for another time ) . I was staying in the capital , | Tokyo |
| WikiBio | Super Mario Land is a 1989 side-scrolling platform video game developed and published by | Nintendo |

Table 4: Example of NOISER attributions on different inputs.

**Gradient-based** methods derive the importance for each token by computing gradients w.r.t. the input (Denil et al., 2015b). The resulting gradient captures intuitively the *sensitivity* of the model to each element in the input when predicting token $w$. While attribution scores are computed for every dimension of input token embeddings, they are generally aggregated at a token level to obtain a more intuitive overview of the influence of individual tokens.

Building upon this, Denil et al. (2015b) takes the input token vector and multiplies by the gradient (Input×Gradient), while Sundararajan et al. (2017b) compares the input with a null baseline input when computing the gradients w.r.t. the input (Integrated Gradients). Nielsen et al. (2022) offers a comprehensive overview of other propagation-based FAs.

**Attention-based** methods are applied to models that include an attention mechanism to weigh the input tokens. The assumption is that the attention weights represent the importance of each token. These FAs include scaling the attention weights by their gradients, taking the attention scores from the last layer, and recursively computing the attention in each layer (Serrano & Smith, 2019; Jain et al., 2020; Abnar & Zuidema, 2020).

**Perturbation-based methods** measure the difference in model prediction between using the original input and a corrupted version of the input by gradually removing tokens (Lei et al., 2016; Nguyen, 2018; Bastings et al., 2019; Bashier et al., 2020). The underlying idea is that removing important tokens will lead the model to flip its prediction or a

significant drop in the prediction confidence. For instance, the input token at position $i$ can be removed, and the resulting probability difference $\mathcal{F}_\theta(X) - \mathcal{F}_\theta(X \setminus x_i)$ can be used as an estimate for its importance. If the logit or probability given to the original output does not change, we conclude that the $i$-th token has no influence. Differently, some perturbation-based techniques utilize a modified model or a separate explainer model to learn feature attributions (Ribeiro et al., 2016; Lundberg & Lee, 2017; Bashier et al., 2020; Hase et al., 2021). LIME (Ribeiro et al., 2016) and SHAP (Lundberg & Lee, 2017) fall into this category.

## 7    Conclusion

In this paper, we introduced NOISER, a perturbation-based input attribution method that employs bounded noise to address the distribution shift problem arising from the discrete nature of text, aiming to explain language model predictions in generation tasks. Furthermore, we proposed *answerability* metrics, a novel automatic plausibility evaluation metric that leverages an LLM to evaluate the relevance of attributed rationales to the target output in the absence of gold rationales or human evaluation. Through comprehensive experiments across three tasks and six LLMs, we demonstrated that NOISER consistently surpasses existing baselines in terms of both faithfulness and answerability rate. Notably, our approach requires no supervision, positioning it as a promising direction for improving model interpretability and efficiency.

## Acknowledgements

Reza was funded by the "*borsa di studio post laurea per attività di ricerca*" from the Department of Computer Science and Information Engineering of the University of Trento. Aryo Pradipta Gema was supported by the United Kingdom Research and Innovation (grant EP/S02431X/1), UKRI Centre for Doctoral Training in Biomedical AI at the University of Edinburgh, School of Informatics. Yu Zhao was supported by the UKRI Centre for Doctoral Training in Natural Language Processing, funded by UK Research and Innovation (grant EP/S022481/1). Pasquale Minervini was partially funded by ELIAI, an industry grant from Cisco, and a donation from Accenture LLP. This work was supported by the Edinburgh International Data Facility (EIDF) and the Data-Driven Innovation Programme at the University of Edinburgh. Gabriele is supported by the Dutch Research Council (NWO) as part of the InDeep project (NWA.1292.19.399). Andrea Passerini was partially funded by the TANGO project, funded by the Horizon Europe Programme, Grant Agreement no. 101120763. Funded by the European Union. Views and opinions expressed are however those of the author(s) only and do not necessarily reflect those of the European Union or the European Health and Digital Executive Agency (HaDEA). Neither the European Union nor the granting authority can be held responsible for them.

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

## A  Soft-NC and Soft-NS Metrics

The Soft-NC and Soft-NS metrics are defined as follows:

$$\text{Soft-C}(X, \hat{y}, X') = \max(0, p(\hat{y} \mid X) - p(\hat{y} \mid X')) \tag{1}$$

$$\text{Soft-S}(X, \hat{y}, X') = 1 - \text{Soft-C}(X, \hat{y}, X') \tag{2}$$

where $X'$ is soft-perturbed versions of $X$ given the following instruction. For the embedding vector $x_i \in X$ and its FA score $s_i$, we modify the elements of $x_i$ using Equation (3).

$$x'_i = x_i \odot e_i, \quad e_i \sim \text{Bernoulli}(q) \tag{3}$$

where e is a binary mask vector of size $n$ (embedding size) and Bernoulli is parameterized with probability $q$:

$$q = \begin{cases} s, & \text{if retaining elements} \\ 1 - s, & \text{if removing elements} \end{cases} \tag{4}$$

The normalized sufficiency and comprehensiveness are then computed using the following equations:

$$\text{Soft-NC}(X, \hat{y}, X') = \frac{\text{Soft-C}(X, \hat{y}, X')}{1 - S(X, \hat{y}, 0)} \tag{5}$$

$$\text{Soft-NS}(X, \hat{y}, X') = \frac{\text{Soft-S}(X, \hat{y}, X') - S(X, \hat{y}, 0)}{1 - S(X, \hat{y}, 0)} \tag{6}$$

However, in generation tasks, the absence of a predictive likelihood for the predicted label makes applying Soft-NS and Soft-NC challenging. Zhao & Aletras (2023) proposed using the Hellinger distance between prediction distributions over the vocabulary as a measure of changes in model predictions. They substitute $p(\hat{y} \mid X) - p(\hat{y} \mid X')$ in Equation (1) with the Hellinger distance. Given two discrete probability distributions, $P_{X,t} = [p_{1,t}, \ldots, p_{v,t}]$ and $P_{X',t} = [p'_{1,t}, \ldots, p'_{v,t}]$, the Hellinger distance is formally defined as:

$$\Delta P_{X',t} = H(P_{X,t}, P_{X',t}) = \frac{1}{\sqrt{2}} \cdot \sqrt{\sum_{i=1}^{v} \left(\sqrt{p_{i,t}} - \sqrt{p'_{i,t}}\right)^2}$$

where $P_{X,t}$ is the probability distribution over the entire vocabulary (of size $v$) when prompting the model with the full-text X. $P_{X',t}$ is for prompting the model with soft-perturbed text. For a given sequence input X and a model of vocabulary size $v$, at time step $t$, the model generates a distribution $P_{X,t}$ for the next token $x_T$. The final Soft-NS and Soft-NC at step $t$ for text generation are formulated as:

$$\text{Soft-NS}(X, x_t, \mathcal{R}) = \frac{\max(0, \Delta P_{0,t} - \Delta P_{X',t})}{\Delta P_{0,t}} \tag{7}$$

$$\text{Soft-NC}(X, x_t, \mathcal{R}) = \frac{\Delta P_{X' \setminus \mathcal{R},t}}{\Delta P_{0,t}} \tag{8}$$

where $\Delta P_{0,t}$ is Hellinger's distance between a zero input's probability distribution and full-text input's probability distribution. $X' \setminus \mathcal{R}$ is the case of "if removing elements" described in Equation (4).

## B    More Results

Table 5 provides the detailed Soft-NS and Soft-NC scores of our experiments.

Table 6 illustrates the answerability metrics on top-50% of feature attributions given the judge model top-5 predictions.

We also analyzed the correlation between the faithfulness score and the answerability metrics on KNOWN in Figure 4. The results show that IG, SHAP, DeepLIFT, IxG, LIME, and Occlusion exhibit similar faithfulness scores, with Occlusion achieving the highest answerability rate and score among them. Meanwhile, NOISER surpasses all baselines in both faithfulness and answerability score but falls short in answerability rate, trailing Occlusion by 5%.

KNOWN

| Method | Qwen2-0.5B | | Llama3.2-1b | | Qwen2-1.5B | | gemma-2-2b | | gemma-2-9b | | Llama3-8b | |
|---|---|---|---|---|---|---|---|---|---|---|---|---|
| | Soft-NS | Soft-NC | Soft-NS | Soft-NC | Soft-NS | Soft-NC | Soft-NS | Soft-NC | Soft-NS | Soft-NC | Soft-NS | Soft-NC |
| Last Attn | 0.0372 | -0.1229 | -0.0301 | -0.0301 | 0.0023 | 0.0584 | 0.1617 | -0.0211 | 0.0321 | -0.3109 | -0.1148 | 0.1052 |
| Rollout | -0.2567 | 0.1406 | -0.0426 | -0.0045 | -0.0264 | 0.1475 | 0.3582 | 0.1042 | 0.0114 | -0.3721 | -0.5818 | 0.3194 |
| SHAP | -0.2714 | 0.7660 | -0.1643 | 0.5388 | -0.0809 | 0.6199 | -0.0050 | 0.3776 | 0.1933 | 0.7270 | -0.3386 | 0.5310 |
| IxG | -0.5373 | 0.7490 | 0.0079 | 0.6980 | -0.0843 | 0.5455 | -0.1152 | 0.6384 | -0.0265 | 1.0541 | 0.0750 | 0.5141 |
| IG | -0.6136 | 0.8312 | -0.1202 | 0.6629 | -0.1209 | 0.6372 | -0.1328 | 0.3343 | 0.1540 | 0.8815 | -0.3386 | 0.7144 |
| DeepLIFT | -0.5430 | 0.8460 | -0.0801 | 0.6274 | -0.0826 | 0.6149 | -0.1296 | 0.4853 | -0.0109 | 0.8747 | 0.0147 | 0.5027 |
| SIG | -0.4989 | 0.5350 | -0.0841 | 0.4375 | -0.0964 | 0.3969 | -0.1494 | -0.0385 | 0.1730 | 0.6147 | -0.1886 | 0.4640 |
| LIME | -0.0221 | 0.2660 | 0.2210 | 0.2920 | 0.0678 | 0.3149 | 0.1336 | 0.2230 | 0.2100 | 0.2732 | 0.5459 | 0.1096 |
| Occlusion | 0.1138 | 0.0489 | 0.2267 | 0.3105 | 0.0606 | 0.1872 | 0.1830 | 0.3510 | 0.2119 | 0.3102 | 0.8302 | -0.0470 |
| NOISER | 0.6785 | 1.5068 | 0.2248 | 1.1741 | -0.0264 | 1.4664 | 0.1805 | 1.2627 | -0.0233 | 2.2001 | 0.8043 | 1.4133 |

LONGRA

| Method | Qwen2-0.5B | | Llama3.2-1b | | Qwen2-1.5B | | gemma-2-2b | | gemma-2-9b | | Llama3-8b | |
|---|---|---|---|---|---|---|---|---|---|---|---|---|
| | Soft-NS | Soft-NC | Soft-NS | Soft-NC | Soft-NS | Soft-NC | Soft-NS | Soft-NC | Soft-NS | Soft-NC | Soft-NS | Soft-NC |
| Last Attn | 1.8502 | 0.0645 | 0.2522 | 0.0733 | 0.0828 | -0.0938 | -0.1583 | -0.0799 | -0.1583 | -0.0799 | 0.0888 | 0.9874 |
| Rollout | 1.8541 | -0.0025 | 0.0894 | 0.1557 | 0.0927 | -0.0126 | -0.1790 | -0.0853 | -0.1790 | -0.0853 | 0.1052 | 1.1231 |
| SHAP | 1.5203 | 2.2767 | -0.3437 | 1.6274 | -0.0323 | 1.6599 | 0.1364 | 1.8382 | 0.1444 | 2.1325 | -0.4277 | 1.1972 |
| IxG | 1.3174 | 2.5798 | -0.1419 | 1.8718 | -0.0682 | 1.6052 | 0.2919 | 2.2883 | 0.2919 | 2.2883 | 0.1390 | 1.9406 |
| IG | 1.5378 | 2.8010 | -0.3141 | 1.6207 | -0.1039 | 1.6537 | 0.3475 | 0.9548 | 0.3475 | 0.9548 | 0.0639 | 3.6551 |
| DeepLIFT | 1.6777 | 2.8214 | -0.0546 | 1.8435 | -0.1606 | 1.7118 | 0.2430 | 2.4998 | 0.2430 | 2.4998 | 0.0657 | 2.0600 |
| SIG | 2.4897 | 1.3749 | -0.3132 | 1.2404 | -0.0297 | 1.1344 | 0.1009 | 0.4404 | 0.1009 | 0.4404 | 0.1895 | 0.9723 |
| LIME | 0.9895 | 0.0870 | 0.1343 | 0.0869 | -0.4042 | -0.0106 | -0.0112 | -0.1524 | -0.0112 | -0.1524 | 0.0923 | 2.2071 |
| Occlusion | 2.2561 | 1.6862 | 0.1370 | 1.8517 | -0.4008 | 1.4153 | 0.5531 | 2.8887 | 0.5531 | 2.8887 | -0.0124 | 4.2364 |
| NOISER | 2.0935 | 4.7119 | 0.3242 | 4.4831 | -0.9588 | 4.1367 | 0.5164 | 3.7563 | 1.1446 | 5.0235 | 0.4705 | 4.6922 |

WIKIBIO

| Method | Qwen2-0.5B | | Llama3.2-1b | | Qwen2-1.5B | | gemma-2-2b | | gemma-2-9b | | Llama3-8b | |
|---|---|---|---|---|---|---|---|---|---|---|---|---|
| | Soft-NS | Soft-NC | Soft-NS | Soft-NC | Soft-NS | Soft-NC | Soft-NS | Soft-NC | Soft-NS | Soft-NC | Soft-NS | Soft-NC |
| Last Attn | 0.8145 | 0.2459 | 0.1041 | 0.5263 | 0.0126 | -0.7181 | -0.1474 | -0.1105 | 0.2246 | 0.0569 | -0.0905 | 0.6405 |
| Rollout | -0.6789 | 0.0385 | 0.0254 | 0.5336 | 0.0424 | -0.7490 | -0.0168 | 0.5253 | 0.2655 | 0.0843 | -0.2114 | 1.0899 |
| SHAP | 0.6624 | 0.8078 | 0.0470 | 1.1202 | -0.0263 | 1.1476 | 0.0071 | 0.7895 | 0.4254 | 2.7240 | -0.0304 | 1.4368 |
| IxG | 1.1577 | 2.2695 | 0.0991 | 1.7374 | 0.2109 | 1.1833 | 0.0320 | 1.5496 | 0.4097 | 2.1950 | 0.2509 | 1.1238 |
| IG | 0.5209 | 1.9008 | 0.0047 | 1.5749 | -0.0232 | 0.7207 | -0.0408 | 1.2318 | 0.5993 | 3.5124 | -0.4283 | 1.1159 |
| DeepLIFT | 0.9232 | 2.2975 | -0.0490 | 1.6755 | 0.0495 | 1.4096 | 0.0574 | 1.4033 | 0.3253 | 1.9753 | 0.1346 | 1.1393 |
| SIG | 1.4616 | 2.3040 | -0.1815 | 1.6114 | 0.3493 | 1.7323 | 0.1572 | 1.2684 | 0.9031 | 4.3250 | 0.6571 | 0.7049 |
| LIME | 1.6490 | 1.3519 | 0.1118 | 0.4537 | 0.1616 | 1.0099 | 0.1260 | 0.5920 | 0.7893 | 2.1634 | 0.3183 | 0.5166 |
| Occlusion | 2.6100 | 2.4950 | 0.3736 | 1.6283 | 0.9353 | 2.9564 | 0.4595 | 2.2637 | 1.3045 | 3.6255 | 1.5924 | 1.7961 |
| NOISER | 4.3005 | 4.4619 | 0.6959 | 3.0427 | 0.9847 | 4.0016 | 0.7941 | 3.4586 | 1.5671 | 5.5837 | 1.4975 | 3.1114 |

Table 5: Soft-NS and Soft-NC Scores Across Datasets.

| Method | Qwen2-0.5B | | Llama3.2-1b | | Qwen2-1.5B | | gemma-2-2b | | gemma-2-9b | | Llama3-8b | | Average | |
|---|---|---|---|---|---|---|---|---|---|---|---|---|---|---|
| | Rate | Score | Rate | Score | Rate | Score | Rate | Score | Rate | Score | Rate | Score | Rate | Score |
| Last Attn | 26% | 0.1722 | 58% | 0.3037 | 21% | 0.1432 | 55% | 0.2917 | 51% | 0.2529 | 48% | 0.2568 | 43% | 0.2368 |
| Rollout | 21% | 0.1445 | 37% | 0.1836 | 23% | 0.1604 | 23% | 0.1086 | 36% | 0.1621 | 40% | 0.1980 | 30% | 0.1595 |
| SHAP | 47% | 0.3904 | 52% | 0.3440 | 47% | 0.3662 | 31% | 0.2188 | 28% | 0.2051 | 43% | 0.2427 | 41% | 0.2945 |
| IxG | 51% | 0.4087 | 58% | 0.4299 | 40% | 0.3057 | 55% | 0.3792 | 47% | 0.3369 | 44% | 0.3062 | 49% | 0.3611 |
| IG | 48% | 0.4004 | 54% | 0.3586 | 43% | 0.3367 | 31% | 0.2308 | 25% | 0.1847 | 42% | 0.2172 | 41% | 0.2881 |
| DeepLIFT | 49% | 0.4138 | 60% | 0.4067 | 40% | 0.3159 | 48% | 0.3271 | 45% | 0.3164 | 44% | 0.2898 | 48% | 0.3450 |
| SIG | 40% | 0.3025 | 44% | 0.2759 | 40% | 0.3091 | 24% | 0.1644 | 36% | 0.2698 | 28% | 0.1584 | 35% | 0.2467 |
| LIME | 49% | 0.3936 | 49% | 0.3315 | 53% | 0.4307 | 61% | 0.4021 | 60% | 0.3916 | 48% | 0.3230 | 53% | 0.3787 |
| Occlusion | 67% | 0.4666 | 61% | 0.4001 | 65% | 0.5093 | 64% | 0.4204 | 63% | 0.4043 | 57% | 0.4241 | 63% | 0.4375 |
| NOISER | 65% | 0.5996 | 58% | 0.5435 | 57% | 0.5474 | 66% | 0.6245 | 60% | 0.5400 | 54% | 0.5962 | 60% | 0.5752 |

Table 6: Answerability metrics on KNOWN dataset w.r.t judge model **top-5** predictions.

## C   Datasets Statistics

Appendix C shows the number of true predictions by each model given KNOWN and LONGRA datasets.

| Dataset | Qwen2-0.5B | Llama3.2-1b | Qwen2-1.5B | gemma-2-2b | gemma-2-9b | Llama3-8b |
|---|---|---|---|---|---|---|
| Known (1208) | 661 | 828 | 774 | 830 | 822 | 875 |
| LongRA (573) | 140 | 160 | 170 | 209 | 148 | 165 |

Table 7: Number of true predictions captured by each model.

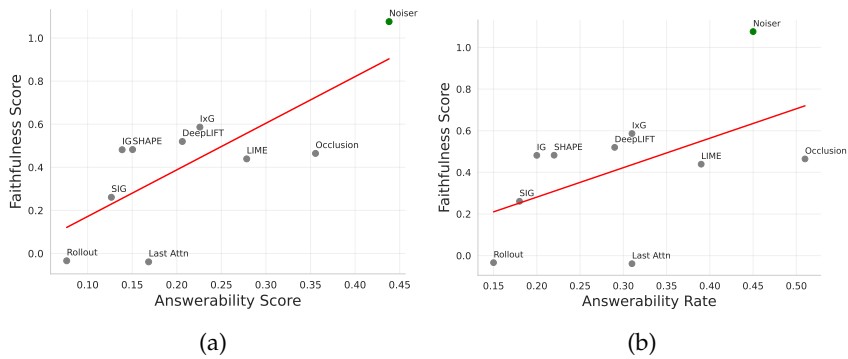

Figure 4: Comparison of average faithfulness score with (a) average answerability score and (b) answerability rate.

# D    Bounding Computations

Since $k_{\min}$ is dependent on the model, we introduce norm-bounding as the norm value of the noise vector $\mathbf{n}$ is different based on the model's embedding size ($d_{model}$). To avoid different norm values for each sample in a given data, we use the expected norm value of the noise vector, $\mathbb{E}\left[\|\mathbf{n}\|_p\right]$, and use $k = \frac{1}{\mathbb{E}\left[\|\mathbf{n}\|_p\right]}$ as the final bounding for the noise vector.

In the following, we show the expected value of each norm given a model with $d_{\mathrm{model}}$ embedding dimensions.

Let $\mathbf{n} \in \mathbb{R}^{d_{\mathrm{model}}}$ be a random vector where each component $n_i \sim \mathcal{N}(0,1)$. Below, we derive the expected values of different norms and compare their properties.

The $L_2$ norm (Euclidean Norm) is defined as follows:

$$\|\mathbf{n}\|_2 = \sqrt{\sum_{i=1}^{d_{\mathrm{model}}} n_i^2}$$

where each $n_i^2$ follows a *chi-squared distribution* with 1 degree of freedom, which results in the following:

$$\mathbb{E}\left[\|\mathbf{n}\|_2^2\right] = d_{\mathrm{model}}$$

By Jensen's inequality and the Law of Large Numbers, for large $d_{\mathrm{model}}$:

$$\mathbb{E}\left[\|\mathbf{n}\|_2\right] \approx \sqrt{\mathbb{E}\left[\|\mathbf{n}\|_2^2\right]} = \sqrt{d_{\mathrm{model}}}$$

The $L_\infty$ norm (Maximum Norm) is defined as follows:

$$\|\mathbf{n}\|_\infty = \max_{1 \leq i \leq d_{\mathrm{model}}} |n_i|$$

The cumulative distribution function (CDF) for $|n_i|$ is $F(x) = \mathrm{erf}\left(\frac{x}{\sqrt{2}}\right)$. The CDF for the maximum of $d_{\mathrm{model}}$ samples is $F_{\max}(x) = [F(x)]^{d_{\mathrm{model}}}$. Using extreme value theory, the expected maximum for large $d_{\mathrm{model}}$ approximates:

$$\mathbb{E}\left[\|\mathbf{n}\|_\infty\right] \approx \sqrt{2\ln d_{\mathrm{model}}}$$

# E    Answerability Evaluation Prompt

Below, we provide the prompt used for evaluating FAs' answerability.

**Answerability Evaluation Prompt**

# Task:
Given a set of words extracted from a prompt for a completion task, return a single word as the most probable completion for the unseen prompt WITHOUT providing any explanation.

