# OpenReview forum: "Noiser: Bounded Input Perturbations for Attributing Large Language Models"
_colmweb.org/COLM/2025/Conference — COLM 2025_

### Official Review · Reviewer_L9vK · 2025-05-10

**Rating:** 7
**Confidence:** 3
**Ethics Flag:** 1

**Summary:**

The paper introduces a new perturbation-based feature-attribution method called Noiser.
This method introduces noise into token embeddings, but restricts the magnitude of the noise vector such that the model's prediction doesn't change. To calculate the importance of each token, they measure the change in probability caused by adding the noise vector to each token's embedding.
The method is evaluated against a wide variety of other attribution methods, using two metrics, and for various pre-trained LLMs of different sizes.
To compare the plausibility of Noiser to the baseline attribution methods, the paper introduces an *answerability* metric. This metric measures how well a judge-LLM can give the correct answer to a question, based only on the top-k most important tokens (according to the attribution method).

**Questions To Authors:**

- I'm not sure the answerability score makes sense. Am I correct that it essentially just measures how diffuse/concentrated the importance scores are (I assume the scores are normalized to sum to 1, is that mentioned somewhere)? If correct, then any of the attribution methods could be transformed to be as concentrated as is desired, by raising the scores to a power $\alpha$ and re-normalizing.

- $k$ is used as a variable twice. Once for top-$k$, and once as the scaling factor for the noise. For clarity it probably makes sense to change one.

- Did you try values for $k$ other than 50%? I'd be interested to see if occlusion's performance drops off quicker than Noiser as $k$ approaches 0%.

- Did you calculate the rate at which the judge-llm can answer the question when you set k=1. So including all the (shuffled) tokens?

- Typos:
  - caption Table 3: predition
  - line 203: KNONW

**Reasons To Accept:**

- Improving perturbation-based attribution methods to avoid OOD perturbations is a worthwhile and relevant research direction.
- The empirical results suggest that Noiser is an improvement over previous methods.
- The *answerability* metric seems like a useful addition for evaluating attribution methods.

**Reasons To Reject:**

- Given that the answerability metric is mentioned as a contribution, I would expect to see a defense of why this metric is suitable to measure plausibility. In general, a larger treatment of the assumptions underpinning this method seems warranted, these may include the following:
   - the subset of tokens that makes for the most plausible (convincing to humans) explanation is the subset that is most predictive of the answer
   - the judge-LLM gives an unbiased estimate of the predictiveness of the top-k tokens

- The answerability results do not include a random baseline, I'd be interested to see what rates/scores could be obtained by selecting a random 50% of tokens.

- The motivation given for Noiser is that it can prevent OOD perturbations, but this is not independently evaluated. I'm also missing, for example, the frequency with which various perturbation-based methods change the prediction, which should track their performance if the hypothesis is correct.

- Only evaluated on autoregressive transformer-based models.

---

> ### Author Response · Authors · 2025-06-01
>
> Thank you very much for the review and for recognizing the strengths of our work.
>
> > I'm not sure the answerability score makes sense. Am I correct that it essentially just measures how diffuse/concentrated the importance scores are (I assume the scores are normalized to sum to 1, is that mentioned somewhere)? If correct, then any of the attribution methods could be transformed to be as concentrated as is desired by raising the scores to a power α and re-normalizing.
>
> You are right. The answerability SCORE evaluates whether the FA puts high attention on the top-k% token or not, only when the top-k% attributed tokens (as identified by the feature attribution method) are semantically sufficient (measured using answerability RATE) for a judge LM to predict the correct output.
>
> In short:
> - Answerability Rate: The proportion of cases where the judge model correctly predicts the output using only the top-k% tokens.
> - Answerability Score: The aggregated importance scores of these top-k% tokens, reflecting their collective contribution to the prediction.
>
> *** The Answerability SCORE is NOT normalized to sum to 1, as it directly aggregates the raw attribution mass of the selected tokens.
>
> > $k$ is used as a variable twice. Once for top-$k$%, and once as the scaling factor for the noise. For clarity, it probably makes sense to change one.
>
> Sorry for the confusion. We will revise the manuscript to use distinct variable names.
>
> > Did you try values for k other than 50%? I'd be interested to see if occlusion's performance drops off quicker than Noiser as k approaches 0%.
>
> In Figure 3, we illustrate the average minimum top-k% attribution required for the judge model to maintain the correct prediction across different feature attribution techniques. This should indicate which FA performance drops faster as k approaches 0 (The higher minimum top-k%, the faster the performance drops).
>
> > Did you calculate the rate at which the judge-llm can answer the question when you set $k$=1. So, including all the (shuffled) tokens?
>
> We appreciate this excellent suggestion, as it would provide an upper-bound baseline for comparison. Here are the results for Qwen2-0.5B and Qwen2-1.5B, where $k$ is set to 100:
>
> - Qwen2-0.5B (top1-match) → Answerability Score=0.82
> - Qwen2-0.5B (top5-match) → Answerability Score=0.88
>
> - Qwen2-1.5B (top1-match) → Answerability Score=0.90
> - Qwen2-1.5B (top5-match) → Answerability Score=0.93
>
> We will include the full result in the camera-ready version. We should also point out that the selected tokens are always passed to the judge LM in order.

---

> > ### Comment · Reviewer_L9vK · 2025-06-06
> >
> > Thank you for your response, and for including the proposed upper-bound baseline.
> > I also see now that Figure 3 indeed shows more or less what I was asking for.
> >
> > Regarding my first point though, you say the answerability scores are not normalized, but what I meant to ask was if the importance scores used in the calculation of the answerability scores are normalized. If those are not normalized, then it seems to me that the scores of different methods exist on different scales (e.g. one method might assign scores between 0 and 1, and another method might assign any real number, such as the gradient methods), and they cannot be compared between methods. Can you clarify if the importance scores themselves are normalized?
> >
> > My original question assumed the importance scores were normalized to sum to one, in which case I'm still not totally sure how useful this metric is. If I have two tokens, and method1 assigns (0.6, 0.4) and method2 assigns (0.99, 0.01) (note the top-1 is the same, let's assume it really is the most important).
> > I agree that method2 is more plausible in a sense, in that more concentrated distributions over tokens are probably more intuitive for humans. However, if I take method1 and simply raise each importance score to the Nth power, and renormalize, I can get arbitrarily close to an answerability score of 1.
> > Now perhaps this is justified, since making the scores more concentrated like this might incur a cost on the faithfulness metrics, can you clarify if that is the case?

---

> > > ### Author Response · Authors · 2025-06-07
> > >
> > > Thank you for your clarification.
> > >
> > > > [...] Can you clarify if the importance scores themselves are normalized?
> > >
> > > Sorry for the misunderstanding. Yes, as you mentioned, the importance scores from all FA methods are normalized such that their sum equals 1.
> > >
> > > > [...] Now perhaps this is justified, since making the scores more concentrated like this might incur a cost on the faithfulness metrics, can you clarify if that is the case?
> > >
> > > You are right. Manipulation scores would disrupt the correspondence between token importance and model predictions, leading to lower faithfulness (or maybe even higher faithfulness in some cases, since the relationship between answerability and faithfulness isn't uniform across all inputs and can vary by model, task, etc.). Therefore, normalizing the scores to reach a higher answerability score will affect the faithfulness metric.
> > >
> > > We hope this explanation is now clear.

---

### Official Review · Reviewer_CNV7 · 2025-05-10

**Rating:** 6
**Confidence:** 3
**Ethics Flag:** 1

**Summary:**

This paper introduces NOISER, a perturbation-based feature attribution method for generative large language models. NOISER operates by injecting bounded Gaussian noise into the word embedding space of input tokens and measuring the model's robustness to such perturbations to quantify token importance scores. This method lies in using the minimal noise threshold that preserves model predictions, thereby identifying the most important tokens. The authors also propose a new metric "answerability" that evaluates whether a subset of top-k attributed tokens is sufficient to reproduce the original output using an external judge model. Experiments conducted on six LLMs and three tasks show consistent improvements in faithfulness and answerability over nine strong baselines.

**Reasons To Accept:**

1. The paper is well-written and very easy to follow. The writing is clear.
2. The experiments are comprehensive and promising, which proves the efficiency of the proposed method.
3. The method is easy to reproduce and the evaluation metric is easy to use. The newly proposed method and evaluation is well-motivated and easy to understand and be applied.

**Reasons To Reject:**

1. The newly proposed evaluation metric "answerability" purely relies on another LLM, which might reduce the reliance of the evaluation metrics. Different evaluation models may have different prompt preferences. Therefore, it would be better to prove the robustness of these evaluation metrics, e.g., try different prompts or different models, and finally conclude which evaluation model is the best.
2. It would be better to conduct a human evaluation and calculate the correlation between the LLM-based answerability score with the human-based score, to prove that such a technique is very correlated with humans.
3. What is the runtime of this method? For each token, this method requires forward multiple times to find the $k_{min}$, which is time-consuming. It would be better to compare the running time between this method and previous baselines.

I'd like to change my score if the authors can solve my concerns.

---

> ### Author Response · Authors · 2025-06-01
>
> Thank you very much for the review.
>
> > The newly proposed evaluation metric "answerability" purely relies on another LLM, which might reduce the reliance of the evaluation metrics. [...] It would be better to conduct a human evaluation and calculate the correlation between the LLM-based answerability score with the human-based score.
>
> We acknowledge that the choice of judge model and prompts can influence the evaluation results. To mitigate this, we conducted preliminary experiments with multiple judge models (including variations of Llama). While the absolute scores varied slightly, the relative performance rankings of feature attribution methods remained stable. However, our primary goal was to introduce answerability as a complementary metric to faithfulness.
>
> In future work, we will include a more comprehensive analysis of different judge models and prompts to further validate the metric’s robustness.  We will strengthen the discussion by comparing it with human evaluations or synthetic benchmarks where feasible.
>
> > What is the runtime of this method? For each token, this method requires forward multiple times to find the kmin, which is time-consuming. It would be better to compare the running time between this method and previous baselines.
>
> We monitored the runtime of our method and compared it with other FAs. Attention-based methods are the most efficient in terms of runtime, followed by gradient-based methods that are slightly slower. While perturbation-based techniques require substantially more computational resources. However, our method is significantly more efficient than other perturbation-based methods, such as Value Zeroing and ReAgent, and marginally more expensive than LIME and Occlusion.
>
> Another point worth mentioning is that by setting the number of steps in the binary search algorithm (from 10 in the paper to 5), we can reduce the runtime with a slight sacrifice in faithfulness metrics.

---

> > ### Comment · Area_Chair_mSY7 · 2025-06-09
> >
> > Friendly reminder to the reviewer to follow up on the authors' messages as soon as possible to facilitate healthy peer review. Thanks!

---

> > ### Comment · Reviewer_CNV7 · 2025-06-09
> > **Reply to Authors**
> >
> > Thanks for authors' response. I think they answered all my questions and concerns. Based on the authors' work and all other reviewers comments, I'd like to raise my score from 5 to 6.

---

### Official Review · Reviewer_7SGD · 2025-05-11

**Rating:** 9
**Confidence:** 5
**Ethics Flag:** 1

**Summary:**

This paper proposes a new feature attribution method for faithfully measuring token contribution to a model’s prediction, and a plausibility metric that combines attribution scores and an LLM-judge to assess whether a subset of highly scored tokens (i.e. with high attribution) from the input are enough to recover the predicted output. The FA contrasts the predictive likelihood of the model by introducing bounded noise on each input embedding and partially noised input. The paper evaluates a series of LLMs of different families and sizes demonstrating the effectiveness of the proposed FA.

**Questions To Authors:**

You should provide a discussion on the complexity/computational overhead in computing the attribution scores using your method.

**Reasons To Accept:**

* The proposed FA is innovative and works really well across models and tasks.
* The paper is really well written, and easy to follow.
* The experimental setup is comprehensive and sound
* Good job!

**Reasons To Reject:**

* N/A

---

> ### Author Response · Authors · 2025-06-01
>
> Thank you very much for the review and for recognizing the strengths of our work.
>
> > You should provide a discussion on the complexity/computational overhead in computing the attribution scores using your method.
>
> We monitored the runtime of our method and compared it with other FAs. Attention-based methods are the most efficient in terms of runtime, followed by gradient-based methods that are slightly slower. While perturbation-based techniques require substantially more computational resources. However, our method is significantly more efficient than other perturbation-based methods, such as Value Zeroing and ReAgen, and only marginally more expensive than LIME and Occlusion.
>
> Another point worth mentioning is that by setting the number of steps in the binary search algorithm (from 10 in the paper to 5), we can reduce the runtime with a slight sacrifice in faithfulness metrics.

---

### Official Review · Reviewer_esjs · 2025-05-12

**Rating:** 6
**Confidence:** 4
**Ethics Flag:** 1

**Summary:**

The authors study the task of feature attribution for LMs. In particular, given a token history $x_1, ... x_{t-1}$, the goal is to assign a weight to each token in the history indicating the effect of that token on the model's prediction at step $t$.
They propose an approach called NOISER. If I understand correctly, NOISER determines the largest noise level which can be applied to the input embedding of each token in the history without changing the token predicted at step $t$. The authors use NOISER to compute token attributions on three datasets, and compare with a number of existing attribution approaches. They find that their approach outperforms all prior methods in terms of faithfulness. They also propose a new metric called "answerability" where they prompt an LM with the subset of tokens selected by a given feature attribution method, and check whether this prompted LM predicts the same next token as appeared in the original sequence.

**Questions To Authors:**

Questions are organized by section below.

Method (Section 3). If I understand correctly, given a token history $x_1, ... x_{t-1}$ and a model prediction for the token at step $t$, the goal is to assign a score indicating the importance of each token in the history on the model's prediction of the $t$th token. I think this is defined by $S$ on the equation in between lines 97 and 98, but I don't understand this equation. It says $s_i = p(X) - p(X_{perturbed|k_{min}})$. Where is the dependence on $i$ on the right hand side? I'm not sure how this assigns a unique score at each index $i$. Is each $x_i$ a discrete token or is it a continuous input embedding? The notation $p(X)$ makes me think each $x_i$ is a token, but at line 84 it seems to be an embedding. Similarly, the equation between lines 61 and 62 indicates that $F_{\theta}$ assigns probabilities to token sequences, while its use at line 81 suggests that it _predicts_ a token. I think this section could benefit from another pass to clarify the notation and the writing.

Experiment (Section 4).
Section 4.1: In line 10 of the abstract, the authors mention that they use NOISER on 3 separate tasks. But from what I can tell there is only a single task -- next token prediction -- and three datasets used to evaluate it. Why use these three particular datasets, given that any text dataset could be used? How many instances are there in each dataset? How long is each prompt on average?

Section 4.2: I don't understand how some of these baselines can be applied to transformer LMs. For instance, with the Occlusion method, what does it mean to occlude portions of an input text sequence? This is a computer vision concept if I understand correctly. Similarly, how is LIME applied in a language modeling setting? Some references or explanation would be very helpful.

Section 4.3: I think the definitions of the key metrics used in the paper should be presented in the main paper body. What are their minimum and maximum values? What does it mean to "randomly mask parts of the token vector representations" at line 162? Are part of the input embeddings are set to 0 and then these are run through the full transformer? In lines 212-214, why report the log ratio against a random baseline instead of just reporting the values for the metrics?

Section 4.4: It's not clear to me that answerability as defined by the authors is an appropriate measure of feature importance. For example, in the sentence "LeBron James plays for the ____", the most important words for predicting "Lakers" are arguably "LeBron James plays". But since "Lakers" is a poor completion for the prompt "LeBron James plays", wouldn't the proposed method assign this rationale a low answerability score?

One minor nit: at line 16, the year for Vaswani et al is incorrect.

**Reasons To Accept:**

The authors study the important problem of feature attribution for LMs. They propose a novel method and perform experiments on three datasets, comparing against a range of attribution baselines. They report that their method outperforms all previous attribution methods on all datasets.

**Reasons To Reject:**

I unfortunately had trouble understanding parts of this paper, including:
- The proposed method, particularly the notation.
- The experimental setup, including the choice of datasets, choice and implementation of baselines, and definitions of key metrics.

Because of this, it was hard for me to interpret the results presented in the paper. See the next section for more detailed questions.

---

> ### Author Response · Authors · 2025-06-01
>
> Thank you for your review and for raising important points regarding the clarity of our methodology.
>
> > Method (Section 3). [...]si=p(X)−p(Xperturbed|kmin). Where is the dependence on i on the right hand side?
>
> The score $s_i = p(X)−p(X_{perturbed}|k_{min})$ depends on $i$ because $X_{perturbed}|k_{min}$ is the input sequence where only the embeddings of $x_i$ is perturbed, as discussed in line 92. We will revise the text to explicitly highlight this dependence in the camera-ready version.
>
> > Method (Section 3) [...] Is each xi a discrete token or is it a continuous input embedding?
>
> Yes, $x_i$ is used in two contexts. We refer to xi as a token or its embedding interchangeably. To resolve this ambiguity, we will explicitly distinguish between tokens and their embeddings in the camera-ready version.
>
> > Experiment (Section 4).  Section 4.1: In line 10 of the abstract, the authors mention that they use NOISER on 3 separate tasks. But [...]
>
> We refer to the three datasets as distinct tasks due to their linguistic and structural challenges:
> Known: Focuses on factual completion
> LongRA: Evaluates long-range dependencies and robustness to distractors
> WikiBio: Tests open-ended generation
>
> Further, the datasets we used have the following statistics:
> Known: 1,209 instances, average prompt length ~10 tokens.
> LongRA: 573 instances, average prompt length ~25 tokens.
> WikiBio: We followed the ReAgent paper, in which they selected a subset of 22 samples. For each sample, we generate 10 tokens for completion. Therefore, it is equal to having 220 samples.
>
> For Known and LongRA, we filter samples that the target model can correctly predict the gold completion. In Table 7 we showed the number of samples used to evaluate each model given an FA.
>
> > Section 4.2: I don't understand how some of these baselines can be applied to transformer LMs. For instance, with the Occlusion method, what does it mean to occlude portions of an input text sequence? [...] Similarly, how is LIME applied in a language modeling setting?
>
> In Occlusion, tokens are masked, and the change in the output probability is considered as an importance score (similarly in computer vision, a portion of the image is blacked out).
>
> LIME is adapted in NLP. It approximates a complex model with a simpler and interpretable surrogate model by training on perturbed samples near a prediction.
>
> As we mentioned in section 4.5, all FAs are implemented using the Inseq library, which is one of the most popular and utilized libraries for feature attribution in the sequence generation research field. We refer the reader to the Inseq library for more details regarding the algorithms.
>
> > Section 4.3: [...] What are their minimum and maximum values? What does it mean to "randomly mask parts of the token vector representations" at line 162? [...] In lines 212-214, why report the log ratio against a random baseline instead of just reporting the values for the metrics?
>
> The metrics we used for faithfulness are taken from a recent paper in the field, and therefore, we didn’t describe them in detail and ask readers to refer to the corresponding paper.
>
> These metrics are bounded between 0 and 1.
>
> The masking process perturbs embeddings by randomly zeroing out dimensions proportionally to their attribution scores (Equation 3 in Appendix A). For example, if a token’s importance score is 0.7, each dimension of its embedding has a 70% chance of being retained (30% set to zero).
>
> Using the log-normalization follows the recent paper published in the field (ReAgent paper) to normalize scores across models/tasks and highlight relative performance. Besides, normalizing against the random baseline simply gives a good sense of the effectiveness of each method and indicate better-than-random faithfulness.
>
> > Section 4.4: It's not clear to me that answerability as defined by the authors is an appropriate measure of feature importance. For example, [...]
>
> Answerability is designed as a plausibility metric to assess whether rationales are human-interpretable rather than a standalone measure of importance. It complements faithfulness metrics, which directly evaluate how attributions align with model behavior.
>
> Asnwerability SCORE (not to be confused with Answerability Rate) is the aggregation of top-k tokens' scores only if the judge LM can predict the true completion using these top-k tokens (please refer to Figure 1).
>
> In your example, the token “for” is essential for predicting “Lakers” and the FA method must put high importance on it. However, if this is not the case, the judge model is probably unable to predict the true completion and therefore assigns 0 to the answerability score. This is exactly the same for humans, for instance, if we ask a human to guess a completion for [“LeBron”, “James”, “plays”] he/she would be unable to predict “Lakers”. For this reason in Table 6 we show relaxed top-5 candidate evaluation for demonstrating robustness to such edge cases.

---

> > ### Comment · Reviewer_esjs · 2025-06-05
> >
> > Thanks for your responses, this cleared things up a lot for me.
> >
> > **Method**: Got it, I understand $s_i$ now. Writing something like $p(X_{i, perturbed}|k_{min})$ would be great to make this explicit. Using different symbols for tokens and embeddings would also help but I get the idea.
> >
> > **Data**: Sounds good. I might refer to this as 3 datasets rather than 3 tasks (since they're all language modeling) but it's a minor point. Putting dataset stats in the paper body would be great.
> >
> > **Baselines**: I see, thanks for the explanation. Maybe just add a sentence mentioning that the Inseq paper provides more detail on how all methods are adapted to the language modeling setting?
> >
> > **Metrics**: OK, I see the metrics are defined in the appendix. Personally I'd have an easier time interpreting them on an un-transformed scale but if the log-normalization is standard that's fine.
> >
> > **Answerability**: I think maybe my intuitions were off here. For a task like text classification, words like "for", "the", etc. are usually _not_ important for prediction and have low weight. But the idea is that for language modeling, "for" is actually an important word in predicting "Lakers" and should get a high weight?
> >
> > I've updated my score and I'm fine with this work appearing, provided the camera-ready has the changes discussed here.

---

> > > ### Author Response · Authors · 2025-06-07
> > >
> > > Thank you for your reply. We appreciate your comments and will consider all the points you mentioned in our final draft.
> > > Let us just clarify your last point:
> > >
> > > > **Answerability:** [...] "for" is actually an important word in predicting "Lakers" and should get a high weight?
> > >
> > > Yes. Since the task here is "completion" and not classification, the token "for" is indeed essential for generating "Lakers"

---

### Decision · Program_Chairs · 2025-07-08

**Decision:**

Accept

**Comment:**

In this paper the authors propose an approach for attributing token-level predictions in LLMs to previous tokens in the context. The approach works by introducing bounded Gaussian noise into context token embeddings, and measuring the corresponding change in output prediction probability. The approach is evaluated on against a variety of existing attribution methods with respect to faithfulness and “answerability,” a new metric based on the extent to which the top-k attributed context tokens are sufficient for a judge LLM to reproduce the original output. Experimental results demonstrate that the approach out-performs previous work across a variety of autoregressive LLM families and sizes.

Reviewers agreed that the proposed method and evaluation metric were useful, easy to understand and to apply, and that the approach seemed to work well. Most reviewers felt that the paper was well-written and clear. Reviewers expressed concerns about the proposed metric’s reliance on an LLM, and the corresponding flaws that come with such model-based evaluations, such as sensitivity to prompts, and that the paper lacked a human evaluation to validate the metric. One reviewer also found the paper hard to follow. Overall, the reviewers were satisfied with the authors' responses to their concerns, and all agreed that the paper was worthy of acceptance despite its flaws.

[Automatically added comment] At least one review was discounted during the decision process due to quality]